# Redox Status in Patients Suffering from Multiple Chemical Sensitivity (MCS): A Pilot Study

**DOI:** 10.3390/jcm14176185

**Published:** 2025-09-02

**Authors:** Paula Aranda-Martínez, Nerea Menéndez-Coto, Ana Coto-Montes, María Martín-Estebané, Germaine Escames, Darío Acuña-Castroviejo

**Affiliations:** 1Centro de Investigación Biomédica, Facultad de Medicina, Departamento de Fisiología, Instituto de Bio-Tecnología, Parque Tecnológico de Ciencias de la Salud, Universidad de Granada, 18016 Granada, Spain; ampaula@correo.ugr.es (P.A.-M.); mmestebane@ugr.es (M.M.-E.); gescames@ugr.es (G.E.); 2Instituto de Investigación Biosanitaria (Ibs. Granada), Hospital Universitario San Cecilio, 18016 Granada, Spain; 3Departamento de Morfología y Biología Celular, Universidad de Oviedo, 33006 Oviedo, Spain; mdezcotonerea.fuo@uniovi.es (N.M.-C.); acoto@uniovi.es (A.C.-M.); 4Instituto de Investigación Sanitaria del Principado de Asturias (ISPA), 33011 Oviedo, Spain; 5Instituto de Neurociencias del Principado de Asturias (INEUROPA), 33006 Oviedo, Spain; 6Centro de Investigación Biomédica en Red de Fragilidad y Envejecimiento Saludable (CIBERFES), Instituto de Salud Carlos III (ISCIII), 28029 Madrid, Spain; 7UGC de Laboratorios Clínicos, Hospital Universitario San Cecilio, 18016 Granada, Spain

**Keywords:** multiple chemical sensitivity, oxidative stress, glutathione, antioxidant enzymes, inflammation

## Abstract

**Background/Objectives:** Multiple chemical sensitivity (MCS) is a complex environmental illness characterized by intolerance to various environmental chemicals, affecting multiple organ systems. Despite its prevalence, MCS remains poorly understood, with limited recognition by the World Health Organisation amid challenges in diagnosis due to symptom heterogeneity. This study aimed to investigate the oxidative stress status in patients diagnosed with MCS compared to healthy controls, focusing on plasma and erythrocyte markers. **Methods**: Blood samples from 40 MCS patients and 40 controls were analyzed for lipid peroxidation (LPO), total antioxidant activity (TAA), adenosine triphosphate (ATP), and antioxidant enzyme activities, alongside glutathione cycle components. **Results**: Results revealed no significant differences in plasma LPO or TAA between groups, with a reduction in 61% ATP levels in MCS subjects. However, erythrocyte analysis showed reduced levels of glutathione (GSH) and total glutathione in MCS patients. Glutathione peroxidase (GPx) activity also decreased by 15% in erythrocytes of MCS patients, suggesting increased hydrogen peroxide detoxification at the expense of oxidation of GSH to glutathione disulfide (GSSG). Because glutathione reductase activity (GRd) did not change, this GSSG could not be reduced, the GSSG/GSH ratio increased by 46%, indicating heightened intracellular oxidative stress. Catalase (CAT) activity also remained unchanged (reduced by 9%, non-significant). **Conclusions**: These findings highlight the role of oxidative stress in MCS pathophysiology, particularly the disruption of the glutathione cycle within erythrocytes. The study underscores the need for further research into the molecular mechanisms underlying MCS to improve diagnostic criteria and therapeutic strategies. Understanding intracellular oxidative imbalances may provide insights into the systemic dysfunction observed in MCS patients.

## 1. Introduction

Multiple chemical sensitivity (MCS) is a disorder of unknown etiology and its prevalence varies widely worldwide, ranging from fractions of a percent (between 0.02% and 0.004% of the population in Spain) to double-digit figures (12.8% medically diagnosed in the USA), depending largely on the criteria and methods employed. Studies based on self-reported questionnaires typically yield higher prevalence rates than those relying on physician-diagnosed cases, reflecting potential reporting bias and the lack of standardized diagnostic criteria. The absence of recognition of MCS by the WHO further exacerbates this heterogeneity, since different countries and research groups adopt divergent definitions. Consequently, both underestimation (due to misclassification or exclusion) and overestimation (due to reliance on subjective self-reports) remain major limitations when interpreting epidemiological data on MCS. MCS primarily affects women, who represent between 60% and 88% of those affected [1]. This disorder appears to correlate with socio-economic factors, as it tends to affect middle-aged women with upper-middle educational levels and high economic status. However, there is no sufficient or clear scientific evidence supporting a direct correlation between MCS and female sex [2,3].

Cullen [4] describes MCS as a syndrome believed to be caused by environmental exposure to various chemicals or toxins, affecting more than one organ and a minority of individuals. It may result from multiple low-dose exposures or a single accidental exposure to high levels of a chemical [1]. Among the most frequent triggers are petroleum derivatives, detergents, solvents, plastics, pesticides, and gases from the textile and paint industries. Lacour et al. [5] expanded on Cullen’s principles and suggested the involvement of the central nervous system (CNS), particularly in relation to olfactory symptoms, which have a social impact and affect the patient’s lifestyle. They also proposed that, to be diagnosed with MCS, the condition must persist for at least six months and involve at least one organ system other than the CNS. Both Cullen’s and Lacour’s criteria are the most widely accepted and applied within the medical and scientific communities. These contributions support the conclusion that MCS is a chronic condition whose symptoms emerge after repeated exposures and tend to resolve or improve when the causative agents are removed from the environment [6].

The heterogeneity of symptoms makes it difficult to understand the molecular basis of this disorder. Classically, individuals with MCS present with fatigue, difficulty concentrating, weakness, malaise, joint pain, and depression, among other common symptoms [7]. In many cases, patients believe the disorder originated in their workplace, which can lead to work incapacity. Although stopping work often reduces symptoms, it may also lead to fear of social reintegration, potentially resulting in renewed social isolation [8].

Patients often present with depression and/or anxiety, which cannot be ruled out as a possible cause of symptoms in at least part of the affected population. It remains unclear whether these psychological symptoms are a cause or consequence of the observed disorder [1,2,9]. However, the potential causes of MCS are so varied that one of the major limitations in studying the condition is the difficulty of establishing a definitive diagnosis. As a result, MCS is often diagnosed by excluding other conditions such as pulmonary, allergic, or autoimmune diseases. Complementary studies, such as olfactometric tests, have shown that patients with MCS tend to have a reduced ability to identify odors, perceive them as more intense, and, in many cases, find them more irritating [8].

Numerous biological theories have been proposed to understand and explain the disorder, including inflammation [10]; dysfunction of the limbic system due to continuous exposure to low doses of chemicals [6,11]; genetic predisposition to increased chemical sensitivity [1]; and oxidative stress, either as a cause or a consequence of impaired natural detoxification processes. Among these hypotheses, oxidative stress has received significant attention due to its mechanistic plausibility and its potential role as a unifying factor linking environmental exposures to cellular damage. For instance, Belpomme et al. [12] and Hirvonen et al. [13] reported elevated levels of reactive species such as nitric oxide (•NO) and peroxynitrite (ONOO−) in MCS patients, suggesting that these could serve as reliable biomarkers of oxidative imbalance. While oxidative stress has been repeatedly implicated in MCS, some large-scale and population-based studies have failed to confirm clear oxidative biomarkers or have reported highly variable results depending on methodology, sample size, and diagnostic criteria.

Although Jacques [14] suggests that MCS and its comorbidities could be treated using psychotherapy and other therapies, the limited understanding of the molecular and cellular mechanisms underlying MCS makes it difficult to implement effective treatments [6]. This therapeutic uncertainty underscores the importance of elucidating the molecular basis of the disorder. Since oxidative stress is one of the most consistently implicated factors in MCS, further investigation into its role could provide valuable insights into its pathogenesis and support the development of targeted interventions. In this context, we consider it worthwhile to deepen our understanding of the oxidative basis of MCS.

## 2. Materials and Methods

### 2.1. Participants

The present study included a total of 40 patients clinically diagnosed with Multiple Chemical Sensitivity Syndrome (MCS) and 40 individuals assigned to the control group, paired by geographical location and age, and recruited through ASESSCA (Asociación de Enfermos de Síndromes de Sensibilización Central de Asturias, Spain). Diagnosis of MCS was established according to Cullen’s and Lacour’s criteria, requiring persistence of symptoms for at least six months, multi-organ involvement, improvement or resolution of symptoms upon avoidance of the suspected agent, and the exclusion of alternative conditions such as allergic, pulmonary, or autoimmune diseases. Features and geographical distribution of the subjects included in the study are shown in Table 1.

The clinical diagnosis of MCS is based on the patient’s history of symptoms and the relationship with the exposure to chemicals and the reactivity to at low concentrations of them. There are no specific laboratory tests or examinations to confirm it; on the contrary, if necessary tests must be performed to rule out other diseases with similar symptoms. The diagnosis follows criteria such suggested by Cullen et al. [4].

All participants previously signed an informed consent form, in which the procedures, characteristics and objectives of the study were specified. The research protocol was approved by the regional ethics committee (CEImPA 2022.508) and was developed in compliance with the ethical principles established in the Declaration of Helsinki.

In all cases, and the subjects (controls and patients) enrolled in this study shared similar features in terms of dietary habits, non-smokers, non-alcohol, and physical activity. The patient’s group treatments are included in Table 1. Controls were not-treated subjects. It should be considered that most of the controls were relatives to the patients (normaly wifes of husbands) and, thus, the habits were also quite similar.

### 2.2. Sample Collection

After overnight fasting, blood samples were collected by venipuncture between 9–10 am from the participants, using BD tubes containing sodium citrate as an anticoagulant agent. Subsequently, the samples were processed immediately upon arrival separating the blood components by centrifugation at 2500× *g* for 15 min. Plasma and erythrocyte fractions were aliquoted for the different analyses to avoid multiple freeze–thaw cycles and stored at −80 °C until further analysis. Quantification of total protein concentration in plasma was performed by the Bradford method [15]. Hemoglobin in red blood cells was determined using Drabkin’s reagent method (Merck) with a spectrophotometer at 550 nm.

### 2.3. Oxidative Stress Studies

#### 2.3.1. Lipid Peroxidation (LPO) in Plasma

The lipid peroxidation process was evaluated following the protocol described by Esterbauer and Cheeseman [16]. This method allows for the quantifying of the formation of lipid peroxides through the measurement of their by-products, Malondialdehyde (MDA) and 4-Hydroxynonenal (4-HNE). For lipid damage measurement, the plasma sample was diluted in 0.9% NaCl buffer (1:3 ratio). The analysis was carried out by preparing a chromogenic reagent, 1-methyl-2-phenylindole, which reacts with MDA and 4-HNE (1,1,3,3-Tetramethoxypropane) for 40 min at 45 °C. After completion of the reaction, the tubes were kept on ice for 15 min and then centrifuged at 10,000× *g* for 5 min at 4 °C. This process yielded a stable chromophore with a maximum extinction coefficient at 586 nm. Results were expressed as nanomoles of MDA + 4-HNE per gram of protein (nmol MDA + 4HNE/g protein).

#### 2.3.2. Plasma Total Antioxidant Activity (TAA)

Total antioxidant activity (TAA) was determined by a spectrophotometric method analyzing the kinetics generated by the interaction of hydrogen peroxide (H_2_O_2_) with ABTS (2,2′-azino-bis(3-ethylbenzothiazoline-6-sulphonic acid)). This reaction, catalyzed by horseradish peroxidase (HRP), produces a highly stable radical with a characteristic absorption spectrum. The procedure used, adapted from the method described by Arnao et al. [17] was performed in a reaction medium containing 50 mM sodium phosphate buffer pH 7.5, 50 mM ABTS, 1 mM HRP and 10 mM H_2_O_2_. This reaction mixture was prepared in advance and kept at 4 °C and protected from light for 4 h before performing the measurement. The total antioxidant activity present in plasma was calculated by measuring the decrease in absorbance at 730 nm after introduction of the sample into the reaction medium (1:24 ratio). The results were expressed as Trolox equivalents (mg/g protein).

#### 2.3.3. Amount of ATP in Plasma

The concentration of adenosine triphosphate (ATP) in the blood samples in plasma was determined by a bioluminescence method. This approach is based on the reaction between ATP and the luciferin-luciferase enzyme complex and allows ATP to be quantified by the emission of light generated by the reaction catalyzed by the enzyme. The analysis was performed using a commercially available ATP bioluminescence kit (FLAA, Sigma-Aldrich, Saint Louis, MO, USA). The protocol followed was that provided by the manufacturer and the plasma samples were diluted 1:50 in H_2_O before analysis. The results were expressed as nanomoles of ATP per gram of protein (nmol ATP/g protein).

#### 2.3.4. Antioxidant Enzyme Activities in Erythrocytes

The erythrocyte fraction was diluted 1:20 in hemolysis buffer and centrifuged at 2500× *g* for 15 min for catalase and GPx assays or diluted 1:5 in water and centrifuged at 10,000× *g* for 15 min for GRd activity. Catalase (CAT) activity was determined spectrophotometrically by monitoring the decomposition of hydrogen peroxide at 240 nm, following the method described by Aebi [18] and expressed as millimolar of catalase per minute per gram of hemoglobin (mmol/min/g Hb). For GPx activity, the supernatant was incubated with either catalyzed or non-catalyzed reaction mix, followed by the addition of t-butyl hydroperoxide, and absorbance at 340 nm was measured for 3 min. For GRd activity, diluted supernatant was incubated with GSSG and NADPH, and absorbance at 340 nm was recorded for 7 min. Both activities were determined from the rate of decrease in absorbance on a microplate fluorescence reader (PowerWaveX; BioTek) due to the NADPH oxidation and were expressed as μmol/min/g Hb [19].

#### 2.3.5. Glutathione Cycle in Erythrocytes

The erythrocyte fraction was diluted 1:1 in hemolysis buffer, deproteinized with 10% TCA (1:1 ratio) and centrifuged at 20,000× g for 15 min. Supernatant was used for GSSG measurement after blocking free GSH with N-ethylmaleimide for 40 min and then diluted in NaOH. GSH samples were prepared directly in phosphate buffer. Both GSH and GSSG samples were incubated with O-phthalaldehyde (OPT) as fluorescent reagent and measured fluorometrically at 420 nm on a microplate fluorescence reader (FLx800; BioTek Instruments Inc., Highland Park, VT, USA), using standard curves and specific dilution factors (0.8 for GSH, 0.133 for GSSG) [20]. The results were expressed as μmol/g Hb.

### 2.4. Statistical Analysis

The data obtained were analyzed and plotted using GraphPad Prism 6 software (GraphPad Software, Inc., La Jolla, CA, USA). The distribution of data in each experiment was assessed for normality using the Shapiro–Wilk test. Depending on the result of this assessment, different statistical tests were used to identify significant differences between the groups compared. Data following a normal distribution (parametric) were analyzed using Student’s *t*-test. In cases where the data did not follow a normal distribution (non-parametric), the Mann–Whitney U test was used. Differences with a *p*-value of less than 0.05 (*p* < 0.05) were considered statistically significant. Results were presented in graphs as mean values ± standard error of the mean (SEM).

## 3. Results

### 3.1. Markers of Plasma Oxidative Stress

Oxidative damage was assessed by measuring lipid peroxidation (LPO) and plasma total antioxidant activity (TAA). The results showed no statistically significant differences between the control group and patients diagnosed with multiple chemical sensitivity (MCS) (82.85 mg Tx/g protein in the MCS group vs. 87.04 mg Tx/g protein in the control group for TAA) or in lipid peroxidation products (127.87 nmol MDA + 4-HNE/g protein in the MCS group vs. 124.14 nmol MDA + 4-HNE/g protein in the control group) (Figure 1).

Plasma ATP levels were lower than expected in both groups (0.92 nmol ATP/g prot in the MCS group vs. 2.37 nmol ATP/g prot in the control group), but no significant differences were observed between the two groups (Figure 2).

### 3.2. Markers of Oxidative Stress in Erythrocytes

The levels of glutathione cycle components are shown in Figure 3. The most significant changes were a reduction in GSH levels (8.44 ± 0.35 μmol/g Hb in controls vs. 7.03 ± 0.27 μmol/g Hb in MCS; *p* < 0.01) and an increase in the GSSG/GSH ratio (1.27 ± 0.04 in controls vs. 1.57 ± 0.05 in MCS; *p* < 0.001) in the MCS patient group. Moreover, GSSG levels also showed a tendency to increase in MCS patients (1.27 ± 0.04 μmol/g Hb in controls vs. 1.47 ± 0.05 μmol/g Hb in MCS), whereas total glutathione (GST = GSH + GSSG) was significantly decreased in the patient group (9.75 ± 0.33 μmol/g Hb in controls vs. 8.16 ± 0.23 μmol/g Hb in MCS; *p* < 0.001).

The activities of GPx, GRd, and CAT in the study groups are shown in Figure 4. GPx activity was significantly reduced in the MCS patient group (56.65 ± 3.60 μmol/min/g Hb vs. 67.85 ± 3.63 μmol/min/g Hb in controls; *p* < 0.05), whereas no significant changes were observed in GRd (1.83 ± 0.26 μmol/min/g Hb in controls vs. 2.10 ± 0.30 μmol/min/g Hb in MCS) or CAT activities (51.37 ± 1.20 mmol/min/g Hb in controls vs. 45.85 ± 1.40 mmol/min/g Hb in MCS).

## 4. Discussion

Although the cause of MCS remains unknown, predictive factors include genetics and epigenetics, psychosocial environment, systemic inflammation, and oxidative stress [21]. MCS has often been inappropriately regarded primarily as a psychosocial condition, which has hindered appropriate treatment for patients. In an effort to address this issue and to clarify MCS diagnosis and treatment, an Italian expert consensus on the clinical and therapeutic management of MCS has been proposed [22]. In the present study, we aimed to explore the presence of oxidative stress in MCS patients. Our results suggest that it mainly affects the glutathione cycle, potentially disturbing the intracellular environment and contributing to the cellular dysfunction observed in these patients [21].

Oxidative stress has been reported in individuals suffering from MCS. Moreover, nearly 40% of these patients exhibit elevated histamine levels, suggesting chronic inflammation and activation of nitric oxide synthase [12,23]. Recently, alterations in SOD2 activity have also been reported in MCS patients [24]. The V16 SOD2 mutation has been associated with reduced plasma total antioxidant activity and decreased glutathione levels in erythrocytes. Impaired SOD activity may hinder the dismutation of superoxide, which, in the presence of inflammation-derived •NO, can lead to the formation of highly reactive peroxynitrite (ONOO−)—a compound also considered an etiological factor in MCS [25]. In this regard, assessing SOD1 and SOD2 activity in MCS patients would be of interest and is planned for future studies.

Our study assessed both extracellular and intracellular oxidative stress status. The glutathione cycle plays a central role in maintaining redox balance. Its reduced form (GSH) neutralizes ROS, thereby protecting proteins, lipids, and DNA from oxidative damage. During this process, GSH is oxidized to glutathione disulfide (GSSG), and the ratio of GSH to GSSG is considered a sensitive indicator of cellular redox status. The glutathione cycle showed significant alterations, suggesting elevated intracellular oxidative stress in MCS patients compared with controls. Additionally, the antioxidant enzymes (GPx, GRd, and CAT) protect cells against oxidative stress, maintaining the effectiveness of cellular antioxidant defenses and contributing to overall redox stability. MCS patients exhibited lower GPx activity than controls, whereas GRd and CAT activities remained unchanged. This selective decrease in GPx activity may indicate either oxidative inactivation of the enzyme or an insufficient compensatory antioxidant response. As a result, reduced glutathione cycle dynamics impair the efficient recycling of GSH, leading to an increased GSSG/GSH ratio. Although GRd activity was unaltered, the imbalance between reduced and oxidized glutathione suggests that GPx dysfunction—together with potentially limited NADPH availability or excessive ROS production—contributes to glutathione system failure.

Although SOD activity was not assessed in this study, it can be hypothesized that at least a slight excess of superoxide production occurs in MCS. The absence of changes in CAT activity suggests that the amount of hydrogen peroxide generated by SOD was low—an interpretation that is consistent with previous studies reporting SOD mutations in MCS patients [24], although this remains a hypothetical explanation in the context of our cohort. Moreover, the reduction in GSH was accompanied by a non-significant increase in GSSG, yet the GSSG/GSH ratio increased significantly. Since GRd activity did not show significant changes, this may reflect the high sensitivity of this SH-dependent enzyme to oxidative damage, as ROS can directly impair its structure [26]. These findings support the hypothesis that GSSG formed during GPx activity was not efficiently recycled to GSH by GRd, which would explain the elevated GSSG/GSH ratio and the resulting intracellular oxidative stress. However, these intracellular alterations cannot be conclusively attributed to MCS alone. Lifestyle and nutritional factors, including diet, smoking status, physical activity, and antioxidant intake, may also influence intracellular redox parameters.

Our study may offer novel insights into the compartmentalization of oxidative stress in MCS, revealing distinct intracellular redox alterations despite unremarkable extracellular markers. Plasma total antioxidant activity (TAA) and lipid peroxidation (LPO) reflect the overall extracellular redox status. TAA represents the overall ability of the bloodstream to neutralize reactive oxygen species, contributing to systemic protection against oxidative damage. LPO indicates the extent of oxidative damage to cell membranes and lipids, reflecting oxidative stress that can affect multiple organs and tissues throughout the body. Our results showed no significant differences in TAA or LPO products in MCS patients compared to controls. This finding is consistent with recent work by Stein et al. [27], who reported preserved systemic antioxidant capacity in MCS patients despite evidence of intracellular redox dysregulation. This dichotomy may reflect adaptive responses to chronic low-grade inflammation. In this context, systemic antioxidant defenses remain functional, whereas intracellular compartments sustain oxidative damage [14,23].

The observed plasma ATP levels, although lower than expected [28], provide relevant pathophysiological insights. Plasma ATP concentrations were reduced in MCS patients compared to controls (Figure 2). While technical factors related to sample processing cannot be entirely ruled out, this pattern resembles findings in chronic inflammatory conditions, where accelerated extracellular ATP hydrolysis occurs due to upregulated ecto-ATPase activity [29]. Notably, extracellular ATP functions as a damage-associated molecular pattern (DAMP), and its rapid clearance may reflect enhanced purinergic signaling in MCS—a mechanism previously implicated in chemical hypersensitivity [21,25].

While our study highlights alterations in intracellular oxidative markers, the overall changes in plasma oxidative stress parameters were modest, suggesting that oxidative stress may play a minor or indirect role in MCS pathophysiology. It is also important to consider the interplay between oxidative stress and inflammation, as inflammatory responses can both induce oxidative imbalance and be triggered by disruptions in redox homeostasis.

MCS presents a unique challenge due to its heterogeneous symptomatology, involving multiple organ systems and varying widely among patients. This variability complicates the identification of shared underlying mechanisms, as it is difficult to determine alterations that are common to all affected individuals despite their diverse clinical presentations [9,27]. In this context, increasing the sample size is essential, as larger cohorts not only enhance statistical power but also enable comparative analyses across populations that differ in climate, diet, and cultural practices—factors known to influence oxidative stress and inflammatory responses [30,31]. However a balance between sample size and the number of variables analyzed must be considered to maintain adequate statistical power.

## 5. Conclusions

In conclusion, the pathophysiological complexity of MCS requires a paradigm shift toward multicenter studies using standardized protocols [30]. Such collaborative efforts are essential to unravel the interactions among genetic, environmental, and psychosocial factors that underlie this complex condition. Extending these investigations to demographically diverse populations may help determine whether oxidative deregulation represents a universal hallmark of MCS or instead a context-dependent phenomenon.

## Figures and Tables

**Figure 1 jcm-14-06185-f001:**
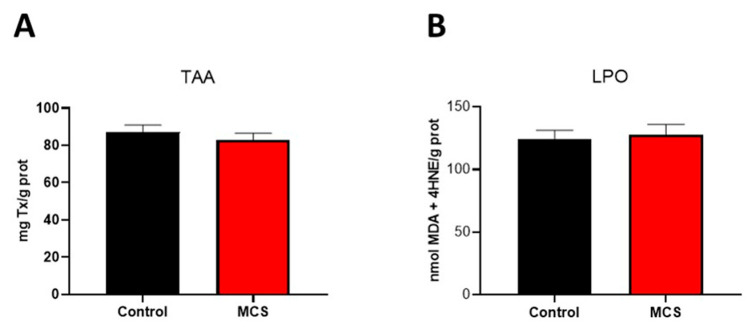
Oxidative stress status in plasma of control individuals and patients diagnosed with MCS. (**A**) Total antioxidant activity (TAA) expressed as mg Trolox/g protein (**B**) Lipid peroxidation (LPO) expressed as nmol MDA + 4-HNE/g protein. Data are represented in the bar graph as mean ± SEM (*n* = 40 per group).

**Figure 2 jcm-14-06185-f002:**
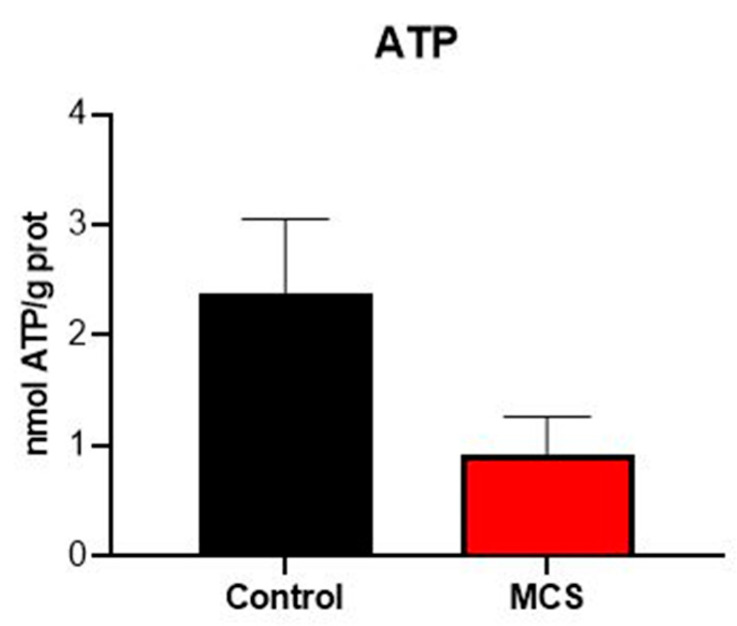
Plasma ATP, expressed as nmol ATP/g protein. Data are represented in the bar graph as mean ± SEM (*n* = 40 per group).

**Figure 3 jcm-14-06185-f003:**
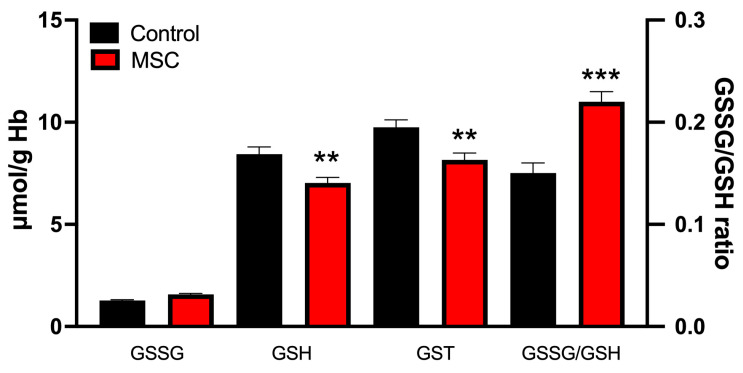
Concentration of GSSG, GSH, total glutathione (GST), and GSSG/GSH ratio in erythrocytes of the subjects studied. Data are presented as mean ± SEM (*n* = 40 per group). ** *p* < 0.01 and *** *p* < 0.001 vs. control.

**Figure 4 jcm-14-06185-f004:**
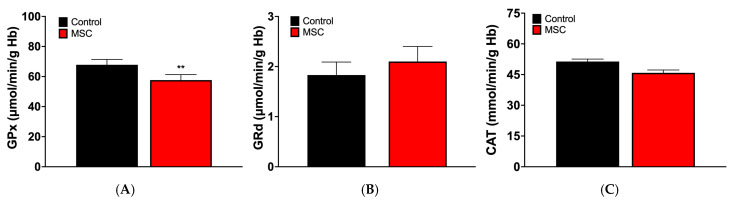
Activities of GPx, (**A**), GRd (**B**), and CAT (**C**) in erythrocytes of the studied groups. Data are presented as mean ± SEM (*n* = 40 per group). ** *p* < 0.01 vs. control.

**Table 1 jcm-14-06185-t001:** Features and geographical distribution of the subjects included in the study.

	MCS	Control	*p*-Value
Age (years)	54.8 ± 7.9	51.4 ± 8.8	
Number of subjects (women)	40 (88.9%)	40 (88.9%)	
Weight (kg)	62.8 ± 12.5	63.7 ± 10.4	
Symptoms	Allergies and Intolerances 19 (42.2%);	Allergies and Intolerances 5 (11.1%)	0.0008
	Chronic Widespread Pain Conditions 9 (20%)	Chronic Widespread Pain Conditions 4 (8.9%)	0.1338
	Thyroid Disorders 4 (8.9%)	Thyroid Disorders 2 (4.4%)	0.3980
	Migraines 4 (8.9%)	Migraines 1 (2.2%)	0.1674
	Gastrointestinal Disorders (Crohn’s disease, gastritis, hiatus hernia, Irritable Bowel Syndrome) 7 (15.6%)	Gastrointestinal Disorders (gastritis, hiatus hernia) 2 (4.4%)	0.0789
	Cardiovascular Disorders (: hyper or hypotension, arrhythmia, atrial fibrillation, sinus tachycardia, extrasystoles) 5 (11.1%)	Cardiovascular Disorders (hyper or hypotension, extrasystoles) 2 (4.4%)	0.2377
	Skeletal and Connective Tissue Disorders 9 (20%)	Skeletal and Connective Tissue Disorders 3 (6.7%)	0.0628
Treatments	Vitamins and Supplements 27 (60%)	Vitamins and Supplements 14 (31.1%)	0.0059
	Cardiovascular Issues (amlodipine, losartan, bisoprolol) 9 (20%)	Cardiovascular Issues (amlodipine, losartan, bisoprolol) 7 (15.6%)	0.5814
	Neurological Medications (fluoxetine, trankimazin, mirtazapine, sertraline) 14 (31.1%)	Neurological Medications (fluoxetine, trankimazin, sertraline) 2 (4.4%)	0.0009
	Anti-inflammatory/Analgesics 2 (4.4%)	Anti-inflammatory/Analgesics 2 (4.4%)	>0.9999
	Thyroid Medications 8 (17.8%)	Thyroid Medications 2 (4.4%)	0.0442
Geographical distribution	Galicia 10 (22.2%), Murcia 10 (22.2%), Asturias 5 (11.1%), Madrid 4 (8.9%), Toledo 4 (8.9%), Ciudad Real 3 (6.7%), Granada 3 (6.7%), Cádiz 2 (4.4%), Cantabria 1 (2.2%), Córdoba 1 (2.2%), Salamanca 1 (2.2%), Ibiza 1 (2.2%)	Galicia 10 (22.2%), Murcia 10 (22.2%), Asturias 5 (11.1%), Madrid 4 (8.9%), Toledo 4 (8.9%), Ciudad Real 3 (6.7%), Granada 3 (6.7%), Cádiz 2 (4.4%), Cantabria 1 (2.2%), Córdoba 1 (2.2%), Salamanca 1 (2.2%), Ibiza 1 (2.2%)	

## Data Availability

The datasets generated during and/or analyzed during the current study are available from the corresponding authors (acoto@uniovi.es; dacuna@ugr.es) upon reasonable request. Materials described in the manuscript will be freely available to any researcher to use them for noncommercial purposes.

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
