# Peer review of "Redox Status in Patients Suffering from Multiple Chemical Sensitivity (MCS): A Pilot Study"

_jcm, 2025, doi:10.3390/jcm14176185_

Round 1

Reviewer 1 Report

Comments and Suggestions for Authors

General Asessment:

The present manuscript addresses a significant yet under-explored subject: the role of oxidative stress in multiple chemical sensitivity (MCS). The authors provide a comprehensive background, employ standard biochemical assays, and discuss intracellular versus extracellular redox alterations. However, while the work is timely and potentially valuable, there are substantial methodological and interpretative weaknesses that must be addressed before the findings can be considered robust and generalizable.

  1. Introduction and Literature ContextThe introduction provides a comprehensive overview of MCS, citing both epidemiological data and potential pathophysiological mechanisms. However, several points require further elucidation and refinement:
    • Epidemiological Uncertainty: The authors cite prevalence estimates for Spain but do not critically address the extreme variability in global prevalence figures or the methodological biases in self-reported versus physician-diagnosed cases. The absence of World Health Organization (WHO) recognition should be more explicitly linked to diagnostic inconsistency and potential over- and underestimation of cases.
    • Pathophysiological Hypotheses – Oxidative stress is presented as a major unifying mechanism; however, the introduction tends to overstate the mechanistic plausibility without sufficiently contrasting contradictory evidence from large-scale studies.
    The Clinical Criteria section provides an accurate description of Cullen's and Lacour's diagnostic frameworks. However, there is a lack of clarity regarding the connection between these frameworks and the current study's recruitment strategy. This lack of clarity leaves the reader uncertain about how strictly these criteria were applied.
  2. Methodology and Study Design
    The methods section is replete with assay protocols but is deficient in critical information regarding patient selection, controls, and confounding factors.
    The study's sample size and power are a matter of concern. With a total of 40 patients and 40 controls, the study may be underpowered for detecting small-to-moderate effect sizes, particularly for biochemical parameters that exhibit high inter-individual variability. The absence of a power calculation is a notable deficiency in the study.
    • Control Matching: While efforts were made to ensure that age and geographical location were matched, other variables—including dietary habits, smoking status, physical activity, and medication use beyond the listed categories—remain unaddressed. These factors have the potential to exert a substantial influence on oxidative stress markers.
    The term "clinically diagnosed MCS" is used without providing any information regarding the clinician's qualifications, the diagnostic tools employed, or the process for excluding psychiatric comorbidities that could potentially confound biochemical results. This lack of detail leads to a lack of diagnostic certainty.
    • Pre-analytical Variability: The authors indicate that samples are to be processed immediately; however, they offer scant information regarding the standardization of collection time, the fasting state of the subject, or circadian influences. It is well-established that these factors can exert a significant influence on oxidative stress parameters.
  3. Results and Statistical Rigor                                                                                            The methods section is replete with assay protocols but is deficient in critical information regarding patient selection, controls, and confounding factors.
    The study's sample size and power are a matter of concern. With a total of 40 patients and 40 controls, the study may be underpowered for detecting small-to-moderate effect sizes, particularly for biochemical parameters that exhibit high inter-individual variability. The absence of a power calculation is a notable deficiency in the study.
    • Control Matching: While efforts were made to ensure that age and geographical location were matched, other variables—including dietary habits, smoking status, physical activity, and medication use beyond the listed categories—remain unaddressed. These factors have the potential to exert a substantial influence on oxidative stress markers.
    The term "clinically diagnosed MCS" is used without providing any information regarding the clinician's qualifications, the diagnostic tools employed, or the process for excluding psychiatric comorbidities that could potentially confound biochemical results. This lack of detail leads to a lack of diagnostic certainty.
    • Pre-analytical Variability: The authors indicate that samples are to be processed immediately; however, they offer scant information regarding the standardization of collection time, the fasting state of the subject, or circadian influences. It is well-established that these factors can exert a significant influence on oxidative stress parameters.
  4. Discussion and Interpretation:                                                                                      The discussion is ambitious but occasionally speculative, extending beyond the confines of the presented data.
    • Intracellular vs. Extracellular Redox Discrepancy – The observed selective changes in glutathione-related parameters are intriguing; however, the authors do not convincingly rule out lifestyle or nutritional differences as explanations.
    • Enzyme Activity Changes – A key finding is highlighted as the reduction in GPx activity; however, the absence of changes in GRd and CAT complicates the mechanistic narrative. The hypothesis of oxidative inactivation is plausible but remains unproven in the current dataset.
    • Speculative Associations with Genetic Variations: References are made to SOD2 mutations and purinergic signaling, yet these associations lack substantiation through genotyping or direct measurement of the relevant pathways within the study cohort. These sections, however, risk an overstated interpretation of causality.
    • Contextualization with Previous Literature – The discussion cites recent work supporting intracellular oxidative stress in MCS; however, contradictory studies showing normal redox parameters in similar cohorts are not discussed. This creates a bias toward a single interpretive direction.
  5. Conclusions and Future Directions: The necessity for multicenter studies and standardized protocols is well-founded. Nevertheless, the manuscript would benefit from a more tempered conclusion, acknowledging the limitations in diagnostic certainty, sample size, and potential confounders. The hypothesis that oxidative deregulation is a "hallmark" of MCS is not supported by the findings from the plasma and erythrocytes, which are mixed.
Comments on the Quality of English Language

Could be improved

Author Response

Thanks you very much for your comments. We addressed them and our comments are below:

  1. Introduction and Literature Context
  • Epidemiological Uncertainty: The authors cite prevalence estimates for Spain but do not critically address the extreme variability in global prevalence figures or the methodological biases in self-reported versus physician-diagnosed cases. The absence of World Health Organization (WHO) recognition should be more explicitly linked to diagnostic inconsistency and potential over- and underestimation of cases.

Clarified in line 41-60.

  • Pathophysiological Hypotheses – Oxidative stress is presented as a major unifying mechanism; however, the introduction tends to overstate the mechanistic plausibility without sufficiently contrasting contradictory evidence from large-scale studies.

Added in line 103-114.

The Clinical Criteria section provides an accurate description of Cullen's and Lacour's diagnostic frameworks. However, there is a lack of clarity regarding the connection between these frameworks and the current study's recruitment strategy. This lack of clarity leaves the reader uncertain about how strictly these criteria were applied.

Added in Materials and Methods, line 129-133.

  1. Methodology and Study Design
  • Control Matching: While efforts were made to ensure that age and geographical location were matched, other variables—including dietary habits, smoking status, physical activity, and medication use beyond the listed categories—remain unaddressed. These factors have the potential to exert a substantial influence on oxidative stress markers.

Clarified in 151-155.

The term "clinically diagnosed MCS" is used without providing any information regarding the clinician's qualifications, the diagnostic tools employed, or the process for excluding psychiatric comorbidities that could potentially confound biochemical results. This lack of detail leads to a lack of diagnostic certainty.

Clarified in lines 129-133.

  • Pre-analytical Variability: The authors indicate that samples are to be processed immediately; however, they offer scant information regarding the standardization of collection time, the fasting state of the subject, or circadian influences. It is well-established that these factors can exert a significant influence on oxidative stress parameters.

Clarified in lines 157-158.

  1. Discussion and Interpretation
  • Intracellular vs. Extracellular Redox Discrepancy – The observed selective changes in glutathione-related parameters are intriguing; however, the authors do not convincingly rule out lifestyle or nutritional differences as explanations.

Added in line372-374.

  • Speculative Associations with Genetic Variations: References are made to SOD2 mutations and purinergic signaling, yet these associations lack substantiation through genotyping or direct measurement of the relevant pathways within the study cohort. These sections, however, risk an overstated interpretation of causality.

Clarify in line 346-350.

  • Contextualization with Previous Literature – The discussion cites recent work supporting intracellular oxidative stress in MCS; However, contradictory studies showing normal redox parameters in similar cohorts are not discussed. This creates a bias toward a single interpretive direction.

This is an important point in MCS research. Due to the variability of the patients, the different types of treatments, and sometimes the different methodologies used to measure the parameters of oxidative stress, it is plausible these differences in the results addressed by the reviewer. For these reasons, we analyzed, as far as possible, a group of patients with similar features as indicated in the lines 151-155.

  1. Conclusions and Future Directions: The necessity for multicenter studies and standardized protocols is well-founded. Nevertheless, the manuscript would benefit from a more tempered conclusion, acknowledging the limitations in diagnostic certainty, sample size, and potential confounders. The hypothesis that oxidative deregulation is a "hallmark" of MCS is not supported by the findings from the plasma and erythrocytes, which are mixed.

Thaks for this information. We agree that a more extensive study with different degrees of MSC and larger number of cases may also improve the conclusions of our study. Is also our interest to include further markers of inflammation including cytokines, prostaglandins, and mitochondrial oxidative stress in a larger, future research.

Reviewer 2 Report

Comments and Suggestions for Authors

In my opinion, this is a well designed study, clearly written and the methods, including the statistical analysis are well conceived and described. However, Table 1 will need some reformatting and additional data needs to be provided:

-units of measure should be added (for instance, kg for weight, years for age, etc.)

-the type of cardiovascular and the other  diseases in subjects and controls need to be elaborated. The same applies for the therapies and the types of supplements used by the individuals taken into the study: they need a more detailed and specific description as it is known that certain therapies or dietary supplements could act as triggers for MCS. The authors should explain if these therapies/supplements/diseases make a difference between the subject and control individuals. The alcohol/drug use and smoking status should also be specified for the subjects and controls.

The abstract mentions 45 controls and 45 subjects while in the body of the text the authors mention only 40 subjects and 40 controls. Which one is true?

Given the rather small number of subjects/controls and some of the missing environmental parameters, this is a pilot study, rather than a fully developed study. I should change the title accordingly: "Redox status in patients suffering from multiple chemical sensitivity (MCS): a pilot study".

English language is adequate, but there are some minor typo errors that need to be corrected, e.g., line 150 "diluted".

Author Response

Thanks you very much for your comments, which has been addressed by us. Please find below our replay:

Table 1 will need some reformatting and additional data needs to be provided:

-units of measure should be added (for instance, kg for weight, years for age, etc.).

Added.

-the type of cardiovascular and the other diseases in subjects and controls need to be elaborated.

From cardiovascular disorders, these included: hyper or hypotension, arrhythmia, atrial fibrillation, sinus tachycardia, extrasystoles.

From gastrointestinal disorders: Crohn's disease, gastritis, hiatus hernia, Irritable Bowel Syndrome.

From neurological disorders: migraines, fibromyalgia,

Others: glaucoma, cystitis, osteoporosis,

The same applies for the therapies and the types of supplements used by the individuals taken into the study: they need a more detailed and specific description as it is known that certain therapies or dietary supplements could act as triggers for MCS. The authors should explain if these therapies/supplements/diseases make a difference between the subject and control individuals.

Although those vitamins and supplements with antioxidative activity were removed from the treatment in all subjects, some of them continue with the followings due to doctor´s indication):

Vitamins and supplements: omega 3, vit B12, vit A, vit D, ashwagandha, Mg, DHA,

Drugs: Eutirox, rivotril, omeprazole, losartan, amlodipine, mirtazapine, rosuvastatin, trankimazin, cidina, fluoxetina, bisoprolol, sertraline,

Suspend treatments 2 weeks before the study: vit C, ubiquinol, PQQ, NAC, glutathione,

Al, the above information was added to the Table 1.

The alcohol/drug use and smoking status should also be specified for the subjects and controls. See lines 151-155.

The abstract mentions 45 controls and 45 subjects while in the body of the text the authors mention only 40 subjects and 40 controls. Which one is true?

Corrected; there are 40+40.

Given the rather small number of subjects/controls and some of the missing environmental parameters, this is a pilot study, rather than a fully developed study. I should change the title accordingly: "Redox status in patients suffering from multiple chemical sensitivity (MCS): a pilot study".

Changed.

English language is adequate, but there are some minor typo errors that need to be corrected, e.g., line 150 "diluted".

Corrected.

Reviewer 3 Report

Comments and Suggestions for Authors

This is a well-written manuscript evaluating the role of oxidative stress in a disease that needs more studies. The manuscript presents several limitations that should be recognized by the authors. For example, the parameters evaluated suggest a minor role of oxidative stress or an indirect effect. The role of inflammatory responses should be better discussed or evaluated, since inflammation can promote oxidative stress and/or be induced by the loss of homeostasis in an oxidative microenvironment. Furthermore, for a broader audience, it is critical to introduce what the parameters (oxidative) evaluated mean for both the cellular and body physiology. Finally, the graphics can benefit from a different format (including the dots and bars at the same time, like in doi: 10.1016/j.freeradbiomed.2021.07.028).

Author Response

Thanks you very much for your comments, which has been addressed by us. Please find below our replay:

The manuscript presents several limitations that should be recognized by the authors. For example:

- the parameters evaluated suggest a minor role of oxidative stress or an indirect effect. The role of inflammatory responses should be better discussed or evaluated, since inflammation can promote oxidative stress and/or be induced by the loss of homeostasis in an oxidative microenvironment.

Added in line 396-401.

- Furthermore, for a broader audience, it is critical to introduce what the parameters (oxidative) evaluated mean for both the cellular and body physiology.

Added in line 331-338, 377-382.

- Finally, the graphics can benefit from a different format (including the dots and bars at the same time, like in doi: 10.1016/j.freeradbiomed.2021.07.028).

Thanks for your suggestion; however, we followed the format of figures in the J Clin Med, and I believe that the figures may remain as they are depicted. However, we can modified them if the reviewer consider it important for the quality of ther manuscript.

Round 2

Reviewer 2 Report

Comments and Suggestions for Authors

The authors have made the necessary corrections, including the one to the title and have properly clarified the issues previously addressed.

Reviewer 3 Report

Comments and Suggestions for Authors

The authors have responded appropriately to all my comments.